# Striving for Stability in the Dough Mixing Quality of Spring Wheat under the Influence of Prolonged Heat and Drought

**DOI:** 10.3390/plants11192662

**Published:** 2022-10-10

**Authors:** Sbatie Lama, Marina Kuzmenkova, Pernilla Vallenback, Ramune Kuktaite

**Affiliations:** 1Department of Plant Breeding, Swedish University of Agricultural Sciences (Alnarp), SE-234 22 Lomma, Sweden; 2Lantmännen Lantbruk, SE-26831 Svalöv, Sweden

**Keywords:** mixing quality, wheat plant, gluten polymers and monomers, dough mixing time

## Abstract

The effects of prolonged heat and drought stress and cool growing conditions on dough mixing quality traits of spring wheat (*Triticum aestivum* L.) were studied in fifty-six genotypes grown in 2017 and 2018 in southern Sweden. The mixing parameters evaluated by mixograph and the gluten protein characteristics studied by size exclusion high-performance liquid chromatography (SE-HPLC) in dough were compared between the two growing seasons which were very different in length, temperature and precipitation. The genotypes varying in gluten strength between the growing seasons (≤5%, ≤12%, and ≤17%) from three groups (stable (S), moderately stable (MS), and of varying stability (VS)) were studied. The results indicate that most of the mixing parameters were more strongly impacted by the interaction between the group, genotype, and year than by their individual contribution. The excessive prolonged heat and drought did not impact the buildup and mixing time expressed as peak time and time 1–2. The gluten polymeric proteins (unextractable, %UPP; total unextractable, TOTU) and large unextractable monomeric proteins (%LUMP) were closely associated with buildup and water absorption in dough. Major significant differences were found in the dough mixing parameters between the years within each group. In Groups S and MS, the majority of genotypes showed the smallest variation in the dough mixing parameters responsible for the gluten strength and dough development between the years. The mixing parameters such as time 1–2, buildup, and peak time (which were not affected by prolonged heat and drought stress) together with the selected gluten protein parameters (%UPP, TOTU, and %LUMP) are essential components to be used in future screening of dough mixing quality in wheat in severe growing environments.

## 1. Introduction

Bread-making characteristics of wheat (*Triticum aestivum* L.) are largely determined by the quantity and quality of storage proteins, and the baking ability is strongly influenced by the genotype and environmental factors [1,2]. Gluten protein is the main determinant of the bread-making quality of wheat, and its content in flour is strongly influenced by abiotic stress factors such as heat and drought. An increase in gluten protein content under drought stress was observed in wheat grain [3]. The stress magnitude was found to vary between the different periods of drought (early, late, prolonged, etc.) [3,4,5], suggesting different consequences for the processing quality of bread wheat flour [6]. 

Dough mixing is a vital step in bread making; it includes blending of the wheat flour with water and developing a three dimensional network of gluten in which the starch granules are embedded [7]. During dough mixing, the rheological properties such as elasticity, viscosity, and extensibility change and these properties are important in predicting the quality of the final product [8], especially under varying growing conditions. The rheological properties of dough are governed by the specific types of gluten protein, such as polymeric glutenins and monomeric gliadins. From these, the unextractable polymeric protein (%UPP) fraction, which can be solubilized in sodium dodecyl sulfate (SDS)-phosphate buffer with the use of sonication, is the most important for gluten quality [7,9,10,11]. Thus, %UPP is known to be directly correlated to the gluten strength in dough and bread volume [10,12,13].

The mixograph is a widely used instrument which mixes wheat flour and water into a dough to assess the processing quality of the wheat flour [14,15,16,17]. Specifically, the mixograph was developed to evaluate mixing characteristics of strong high-protein flour [18,19]. Different versions of the mixograph are available, such as mixographs with 2, 5, 10, and 35 g of flour [18], where the 10 g mixograph is more commonly used [12,14,19]. From the mixograph parameters, the peak time and buildup (the difference between the maximum stress in the dough during deformation and the stress in the dough at the point in time when all the liquid has been absorbed) were found to be positively correlated with the SDS-unextractable polymeric proteins (i.e., gluten strength) [12,20,21] and bread volume (correlation of more than 80%) [19]. However, there have been only few studies conducted using a mixograph to evaluate the processing quality of wheat flour under varying environmental conditions, and none which involve extreme heat or drought stress. 

An extreme stress environment can be defined as a growing environment in which heat or drought stress (or a combination of these) occurring around plant flowering, which is known to cause severe losses in yield and quality. Few studies have indicated that heat and drought stresses induce the formation of large gluten polymers (i.e., %UPP) and overall increase the gluten strength of wheat flour [22,23]. This trend was also observed in our latest study involving wheat flour from genotypes grown during excessively long period of heat and drought [5]. For wheat flour dough, it was reported that a relatively high temperature (e.g., 27 °C) caused a longer dough development time and higher dough stability, resulting in greater loaf volume compared to cooler temperatures (e.g., 18 °C) [24,25]. In particular, heat stress was found to increase both the extensibility and strength of wheat dough [26,27]. Drought was found to increase the optimum dough mixing time [28]. Still, it is unknown how severe heat and drought stresses impact wheat dough characteristics and whether any of these characteristics could be used in the screening of wheat material for climate stability. So far, no studies have focused on the combination of extreme heat and drought stress and its impact on the rheological properties and gluten protein characteristics of the dough of Swedish bread wheat; thus, a major knowledge gap still exists.

The main goal of this study was to investigate the effect of the combination of extreme heat and drought stress—observed in 2018 in a field in Sweden—on the mixing properties and composition of gluten proteins in wheat dough, studied by mixograph and size exclusion liquid chromatography (SE-HPLC). The novel approach of this study is exposing of wheat genotypes to an extreme growing environment: a temperature rise of 6–11 °C above the average temperature and a very low level of precipitation (below 30 mm), and evaluating how the mixing parameters vary in three groups of spring wheats with diverse gluten strengths. The results of this paper highlight how the specific dough mixing parameters, if tuned with the gluten protein characteristics, might be the key criteria to achieve more uniform bread baking performance under varying growing conditions in the future.

## 2. Results

### 2.1. Effect of Year and Groups on the Dough Mixing Characteristics 

Analysis of variance (ANOVA) results show that the interactions of group × genotype, group × year, and group × year × genotype significantly (*p* < 0.001) influenced all the mixing characteristics from the water absorption and dough development phases and IHTP, except the peak width and breakdown (Table 1). The most highly significant (*p* < 0.001) impact from the group × genotype interaction was on IHTP, followed by the group × year × genotype impact. 

The strongly significant impact of year (*p* < 0.001 and *p* < 0.05) was clear on almost all of the mixing parameters, except time 1–2, buildup, and peak time (Table 1), indicating the stability of these parameters in the studied years. The year impact on IHTP was much weaker (*p* < 0.05) compared to the impact of the group (*p* < 0.001) (Table 1). The group showed a clearly significant impact (*p* < 0.001) on time 1–2 and the main dough development phase parameters (peak time, peak width, and build width) (Table 1). 

From the Tukey test, the major significant differences between the C and HD years were found for the initial slope and initial width among S and VS groups, and time 1–2 for M group (Table 2). In Group S, the strongest impacts of the different years were for peak height, width built, IHTP, and water absorption (Table 2). The HD year had a stronger impact only on width build in this group (compared with group S-C). Meanwhile, in group MS, the HD year showed a greater impact on time 1–2 and peak time compared to the C year. No impact of HD year was noted in Group VS, indicating that the C year caused higher values for initial slope and initial width (Table 2).

### 2.2. Variation in Dough Mixing Characteristics among the Groups

The subtracted values between the C and HD years were compared between the groups to show data intervals and to refer this to lower variability (eventual stability). The mixing parameters responsible for dough development and water absorption (peak time, initial width, initial build width, and water absorption) are shown in Figure 1. The studied groups showed similar data distribution between the years, however, some minor differences were observed for the selected mixing parameters. Smaller data distribution intervals between the years were observed for the peak time and initial build width for Groups S and MS compared to Group VS (Figure 1A,C), indicating lower data variation between the years. The different years did not impact the initial width among the groups, although a greater data variation interval was found for Group VS compared to the other groups (Figure 1B). Water absorption is an important dough mixing parameter largely dependent on gluten protein properties and amounts, and the data distribution was found to be similar between Groups S and VS (Figure 1B,D). This indicates that genotypes with varying mixing stability are present across all groups in this study. In conclusion, Groups S and MS are the main groups that offer genotypes with less variation in some of the mixing properties, while Group VS includes genotypic material with mixing properties that vary more broadly.

### 2.3. Relationships between the Mixograph Parameters and the Gluten Proteins 

PCA analysis was performed to investigate relationships between the mixing characteristics of dough and the gluten protein characteristics of flour (Figure 2). PCA results showed that PC1 and PC2 explained 27.9% and 17.1% variability, respectively (Figure 2). The gluten protein parameters describing large polymeric proteins (%UPP, TOTU, %LUPP) and large monomers (%LUMP) were most closely associated only with buildup and water absorption, parameters that are strongly related to protein content of the flour (Figure 2). The mixing parameters such as end height, time 1–2, peak time, and IHTP were those more closely associated with the large polymeric and monomeric proteins. 

The protein content indicating parameters such as, total SDS-extractable proteins (TOTE) and ratio of monomers to polymers (Mon/pol) were positioned in the opposite direction of %UPP, %LUPP, total SDS-unextractable proteins (TOTU) and %LUMP, showing a negative correlation between the parameters compared. Initial slope and initial width were the most closely positioned mixing parameters to TOTE and Mon/pol, followed by break down, peak width, and end width (Figure 2). Total amount of monomeric proteins (TMP) was the most closely related to the mixing parameters responsible for dough development, such as area within, initial build width, initial buildup, and peak height.

From Spearman’s correlation analysis, the highest significant correlations were found between the water absorption and TMP (0.79), TOTE (0.59), TOTU (0.59), and Mon/pol (0.49) (Figure 3). The mixing parameters, buildup, end height, and peak height showed significant correlations of 0.39, 0.43, and 0.42 with protein content, respectively. The highest negative significant correlations were observed between the initial width and %LUMP (−0.46) and %UPP (−0.43) (Figure 3). 

### 2.4. Variation of Monomeric and Polymerics Proteins in Dough under Diverse Growing Conditions

The diverse growing conditions of the studied years impacted the monomeric and polymeric gluten proteins in dough differently, as shown by SE-HPLC (Figure 4). Heat and drought stress resulted in lower values in both TMP and TPP in most of the samples from all groups. The least variation in TMP between the wheat genotypes and between the years accounted for around 50% of the studied genotypes in Group S (Figure 4A). A great number of genotypes from Group S showed similar TPP values between the years (indicating robustness to prolonged heat and drought) as compared to Groups MS and VS (Figure 4B). Several genotypes from Group VS showed rather similar TMP and TPP values in both studied years, showing certain robustness potential to the contrasting growing conditions of these genotypes. 

The amount of polymeric proteins (%UPP) in dough was compared between the diverse groups and years, and higher values were observed for the majority of wheat genotypes from the HD year as compared to the C year in all the groups (Figure 5). The highest %UPP in dough was around 15% for several wheat genotypes from Groups S and MS. The wheat genotypes that were least impacted by the different years (under 3% difference) were 1, 3, 4, 5, 7, 13, 14, and 18 from Group S; all the genotypes from Group MS; and two genotypes (24 and 26) from Group VS (Figure 5). From these, the genotypes 3, 13, and 14 were those that also showed similar TMP and TPP values compared with other genotypes in this study (Figure 4). 

## 3. Discussion

### 3.1. Environment-Induced Changes in Dough Mixing Quality

The impact of heat and drought stress on wheat dough quality has been vaguely investigated, with the few studies performed so far indicating roughly equal impacts from the wheat genotype and the growing environment [28,29]. In the present study, we had unique excessive growing conditions for the Nordic climate, such as prolonged drought and heat in 2018, which positively affected the gluten strength in flour for a number of genotypes [5], and a similar trend was observed in dough for some of the mixing parameters of the individual genotypes. In this study, interestingly, we observed that the excessive heat and drought (i.e., the year) did not impact the two mixing parameters responsible for dough strength and development (time 1–2 and peak time). This observation was different to the previously reported trends such as heat stress induced dough weakening [30,31]. A possible explanation is that differential changes occurring in the size and amount of polymeric proteins are negatively compensated by the lower protein content in the excessive growing environment [5]. 

It is important to point out that, in this study, the interaction between the wheat genotype, year, and group made a larger impact on the dough mixing parameters than the group and year individually, confirming that the environmental events are an important source of variation. This was also previously found for the dough mixing time, which was found to be largely influenced by the interaction between the wheat genotypes and environment [32]. In fact, in our study, the strongest impact from genotype × environment was observed on IHTP, which can be used as a dough strength and development indicator, and in several studies this parameter has been shown to have a good correlation with bread volume. Previous studies have also shown that buildup and peak time are closely linked to the gluten strength [12,21,33], while water absorption is highly correlated with the protein content [18], a trend that was also shown in our study. The peak time and buildup were among those mostly related to the polymeric and large monomeric gluten proteins (i.e., %UPP, TOTU, %LUPP, and %LUMP). Furthermore, time 1–2, peak time, peak width, and IHTP were parameters strongly correlated with the group and genotype. Another aspect strengthening the impact of the interactions group × genotype and group × year involves a clear impact of the genotype in this study. Our results align well with the concept that dough optimum mixing time and torque are parameters controlled by a strong genetic factor [29]. This also explains why the year in this study did not impact time 1–2 or peak time. Since dough mixing time is an indicator of the wheat flour strength [34] and peak time is known to be controlled by the gluten protein composition [35,36], it seems that mixing time and time 1–2 are parameters that might have some potential in wheat genotype prediction in excessive growing environments. However, there remains a key point to understand: where is the important switch putting genetics over environment and vice versa? Some of the water absorption phase parameters, such as initial width and initial slope, were closely related to protein content (as TOTE) in this study. Indeed, polymeric glutenins are known to correlate positively with peak time, while monomeric gliadins show negative impact and weakening of dough [37,38], overall, the protein content decreased in the excessive heat and drought year. Sufficient hydration of wheat flour particles and proteins therein facilitates optimum gluten development during mixing [39]. From the difference in water absorption between the years, which decreased considerably between the groups during the prolonged heat–drought year and varied mostly within the groups in the different years (Table 2), it can be assumed that gluten protein qualitative and quantitative components differed greatly between the years. The higher protein content of flour and weaker gluten in the cool year and an opposite trend for heat and drought stress suggest that water absorption and mixing time might not be optimal for such flours and should be further explored. Extreme mixing (i.e., overmixing) to evaluate hydration of the flour and individual tuning of water amount according to gluten strength should be further studied and not overlooked. Similar findings were also observed in wheat flours used in noodle dough by Liu et al. [40], where flours with different protein contents and gluten strengths behaved differently during mixing.

### 3.2. Growing Conditions-Induced Variation between the Groups 

In our latest study [5] and a few other studies [28,29], we pointed out that either heat or combined heat and drought stress favor gluten strength. When comparing the differences between the years and the groups in this study, we observed this favoring trend for a great number of mixing parameters in the studied groups (initial slope, initial width, initial buildup, initial build width, peak height, width build, IHTP, and water absorption in Group S; time 1–2, peak time, and water absorption in Group MS; and initial slope and initial width in Group VS (Table 2). In general, we did not see clear differences in the mixing characteristics between the groups for most of the mixing parameters in this study, except the peak time and water absorption (Figure 1), most likely due to the fact that differentiation was mainly based on the polymeric proteins that were used in PCA differentiation and grouping. A reason for this choice was based on Swedish baking industries’ needs for high-gluten strength flour, which is further used as fortifying “material” in blends with weaker wheat flours. The produced wheat flour blends are further processed into different types of bread and other wheat products, which is a common practice in Sweden and other countries in Europe. 

The clear impact of heat and drought stress on most of the mixing parameters observed for Group S, which differed for Groups MS and VS, could be explained by higher variation in %UPP in the majority of wheat genotypes and the larger size of Group S. The sensitivity of certain wheat genotypes from Group S to heat and drought stress (such as nr 7) led to a decrease in %UPP in the flour from 66.91% to 61.86%, which could also be one of explanations for observed differences in the dough. Smaller variation in peak time and initial buildup in Group MS indicates similar dough development and water absorption patterns, and these parameters can be suggested for further screening tests. 

Heat and drought reduced the glutenin content (and moisture of flour) in this study, indicating a similar tendency observed in other studies [41,42]. However, this trend is rather uncommon when compared with a significant rise in grain protein due to heat, which was reported in several other studies [30,43,44].

### 3.3. Relation between the Mixing Parameters and Protein Composition

Mixograph parameters are important in predicting bread baking performance, and have been shown to around 90% correlate to bread volume. The bread baking process is a time- and resource-consuming process, and if well correlated with protein composition, can be a very useful tool in breeding. Among mixing parameters, the greatest positive correlations (0.79 and 0.59) were observed between the water absorption and the monomeric gluten proteins (i.e., TMP and TOTE) and polymeric proteins (i.e., TOTU) (Figure 3), which suggests a close relationship between the studied parameters; as such, they might potentially be further used in flour and dough prediction studies. Negative correlations (−0.43 and −0.46) between the initial width and initial slope and gluten strength (i.e., %UPP) as well as %LUMP shown in this study can be explained by the decrease in protein content. More thorough correlation studies taking into account repetitive measurements of the relationships between the mixing and protein composition parameters should be performed. Besides the initial experiments we performed in this study, water absorption, TMP, TOTE, and TOTU may be valuable parameters for investigating genotype selection under diverse growing conditions.

In this study, we observed differences in the gluten protein composition of dough between the years for the total polymeric protein fraction (TPP) (Figure 5), which confirms different genotypes’ sensitivity to the excessive growing environment originating from genetic makeups [26]. It is important to point out that much lower %UPP values were observed in dough in comparison to those in flour for all the groups [5]. This indicates that mixing action might have been insufficient to contribute to the optimum development of a protein network, where a matrix of glutenins and gliadins develops. The monomeric and polymeric types of gluten from the heat and drought year were larger in size and complexity than those from the cool year, as was observed in flour by Lama et al. [5]. Meanwhile, in dough, these differences were expected to be more pronounced, and were possibly the main reason for very different strengths of gluten networks under the studied growing conditions. Different mixograph dough development curves in terms of shape were observed between the years for more than half of the studied wheat material. Width of the mixograph curve can commonly be related to dough extensibility and mixing tolerance, whereas height of the curve represents dough strength and consistency [45,46]. In fact, in this study, initial slope and initial width were negatively correlated with %UPP, %LUPP, and %LUMP, indicating a negative relation with dough extensibility. A negative correlation between the gluten polymeric proteins (LMW-GSs) and dough extensibility during severe heat and drought stress was also found by Phakela et al. [47]. 

In breeding, screening of wheat genotypes according to mixing properties and gluten protein properties (e.g., %UPP and other characteristics of protein) is very important, and this study reveals a number of parameters that might be important in varying growing environments, including excessive heat and drought. However, it should not be forgotten that not only flour and dough, but also bread properties should be considered and evaluated in further techno-functional studies of wheat plant materials. To conclude, in this study, the key parameters to consider in further investigations and screening are time 1–2, peak time and water absorption. These parameters are known for their ability to retain gas in the dough during proofing and baking, and thus are related to the bread volume [48]. Other parameters such as buildup, initial slope, and initial width might also be important in selecting wheat genotypes for less-variable mixing quality. The mixing parameters’ screening and tuning should be performed in relation to qualitative and quantitative gluten protein characteristics such as TOTE, TOTU, %UPP, %LUPP, and to some extent Mon/pol in both flour and dough.

## 4. Materials and Methods

### 4.1. Plant Growing Environment 

This study is an extension of our previous study on 294 spring wheat genotypes grown in 2017 and 2018 [5]. In this study, 56 spring wheat genotypes were used in this study to investigate dough mixing characteristics. The growing seasons in 2017 and 2018 were designated as cool (C) and prolonged heat–drought (HD), respectively. The extreme prolonged HD season was designated due to the higher temperature (6–11 °C higher than the average) and its unusual length lasting from May until grain harvesting in August 2018 (Figure 6). 

The genotypes were grown in a lattice design with two replicates in the C year and a modified augmented design 2 (MAD2) with one replicate in the HD year. Out of two replicates produced, plants from one replicate were used for the C year in order to compare with the HD year. Specific details regarding the growing conditions of the plants in the C and HD years are included in Lama et al. (2022). 

### 4.2. Flour Materials 

The flour of 56 genotypes used in this study were divided into three groups according the variance of the gluten protein parameters %UPP, TOTE, TOTU, %LUPP, and Mon/pol measured by SE-HPLC [12,13] in C and HD environments (data obtained in our previous study by Lama et al. 2022) in PCA analysis. Three PCA distance intervals were designated as follows: (1) stable (S) group (31 genotypes; interval 0.17–1.42; %UPP ≤ 5%), (2) moderately stable (MS) group (13 genotypes; interval 1.45–2.57; %UPP ≤ 12%), and (3) varying stability (VS) group (12 genotypes; interval 2.58–9.10; %UPP ≤ 17%).

### 4.3. Dough Mixing Using Mixograph 

The whole wheat flour (10 g) of 56 genotypes from the two years was mixed with water into a dough using a mixograph (Bohlin Reologi AB, Lund, Sweden), and mixing was performed at 26 °C [49]. Each flour sample was mixed for 10 min in order to determine the optimum mixing time. Two replicates of dough mixed to the optimum time were used in this study. The dough samples after mixing were stored immediately at −80 °C and freeze-dried (Cool safe Pro, LaboGene, Denmark) afterwards for 48 h. Freeze-dried dough samples were ground into a fine powder using a grinder (Yellow line, A10, IKA-Werke, Staufen, Germany) and used for further analysis. 

Seventeen dough parameters were obtained from mixograph curve for the wheat genotype Mirakel grown in the C year (Figure 7). The parameters are (1) initial slope (A1/T1), (2) initial width (A1−B1), (3) initial buildup (A2−A1), (4) time 1–2 (T2−T1), (5) initial build width (A2−B2)−(A1−B1), (6) buildup (A3−A2), (7) peak time (T3), (8) peak width (A3−B3), (9) peak height ((A3+B3)/2), (10) width build (A3−B3)−(A2−B2), (11) break down (A3−A4), (12) end width (A5−B5), (13) end height ((A5+B5)/2), (14) area below (A1−A5), (15) area within (area between A1−A5 and B1−B5), (16) IHTP (integrated height to the peak), and (17) water absorption (obtained according to Wikström and Bohlin [14]). The process of mixing dough was divided into three phases, designated as water absorption (parameters 1–5), dough development (parameters 6–10), and break down of dough (parameter 11) [14,50]. Overall, 13 mixing parameters were used for evaluating the effect of year and groups, and 17 mixing parameters were used for Spearman’s rank correlations between the protein parameters studied by SE-HPLC and flour protein content determined by NIT; gluten protein and flour parameters borrowed from Lama et al. (2022) were used in this study for Spearman’s rank correlations. 

### 4.4. Size Exclusion High-Performance Liquid Chromatography of Doughs (SE-HPLC)

Thirty genotypes were selected for gluten protein analysis of dough by SE-HPLC. The selected wheat materials comprised 18 genotypes from Group S, 4 genotypes from Group MS, and 8 genotypes from Group VS.

Freeze-dried dough samples were blended with the buffer and two step extractions were performed to investigate the gluten protein polymerization (extractability) in dough according to Lama et al. [5] and Kuktaite et al. [12,51], with some modifications. The modifications were as follows: after first extraction (1Ex, referred to as SDS-extractable protein) and second extraction (2Ex, referred to as SDS-unextractable protein), the supernatants were collected in SE-HPLC vials and heated at 80 °C for 2 min (to inactivate proteases) in a water bath according to Islas-Rubio et al. [52]. Immediately after heating, the vials were cooled down in ice cold water for 1 min, followed by SE-HPLC analysis. 

We injected 20 μLof extracted proteins from 1Ex and 2Ex into an SE-HPLC column (BIOSEP SEC-4000 Phenomenex column), which were separated for 30 min in a solution of 50% acetonitrile with 0.1% trifluoroacetic acid (TFA). The extracted chromatograms at 210 nm UV wavelength were divided into four areas according to the retention times of different gluten proteins. The gluten protein parameters of TOTE, TOTU, %UPP, %LUPP, percentage of large unextractable monomer into total large monomer (%LUMP), and mon/pol were calculated according to Lama et al. [5]. Total polymeric proteins (TPP) and total monomeric proteins (TMP) were calculated as LPP+SPP+LPPs+SPPs and LMP+SMP+LMPs+SMPs, respectively.

### 4.5. Statistical Analysis 

The statistical analysis was performed using the software R (https://www.r-project.org/). Three way ANOVA was conducted for evaluating the impact of the group (S, M, and VS) and year (C and HT), and the interactions of group × genotype and genotype × year × group on the mixing parameters (for this analysis, “wheat genotypes” were nested within the “group”; since each wheat genotype belongs to exactly one level of group, group and wheat genotype effects were not differentiated). For water absorption (ml/10 g flour) the values did not differ between the replicates and therefore average value was used for two way ANOVA analysis. Tukey’s post hoc test, PCA, and Spearman’s rank correlation were performed to study the variation of dough mixing parameters in the C and HT years. 

## 5. Conclusions 

Striving for stability of wheat quality characteristics in varying and excessive growing environmental conditions is essential in wheat breeding programs. From this point of view, there is a continuous interest to define wheat quality parameters that are most reproducible and are more influenced by genotype than by growing conditions. In this context, from our study, there was a significant impact of the year on most of the dough mixing parameters, except time 1–2, buildup, and peak time. These parameters were strongly impacted by the genetic background and might be very useful in the screening of wheat material from contrasting environments. Therefore, in screening procedures of wheat breeding activities, the interaction of genotype × environment should be thoroughly explored. 

The varying growing conditions were the main factor causing differences in the dough mixing parameters within the studied groups (S, MS, and VS) and minor differences between the groups. In Groups S and MS, the majority of wheat genotypes showed less variation in dough mixing characteristics such as peak time, initial width, initial build width, and water absorption, which could be related with the gluten strength and dough development. Group VS included genotypic material which broadly varied in the mixing characteristics.

The gluten protein parameters for the large polymeric proteins (%UPP, TOTU, %LUPP) and the large monomers (%LUMP) in the flour showed a close association with the buildup and water absorption in dough, indicating their potential to be used as screening parameters for wheat dough stability. However, further studies are needed to better fine tune those gluten protein parameters for diverse excessive growing environments and different dough mixing conditions (e.g., overmixing and optimal hydration).

To sum up, screening of wheat genotypes according to the dough mixing characteristics and the gluten protein parameters (e.g., %UPP and others) is very important, and this study reveals a number of parameters that might be important to focus on in contrasting growing environments. However, screening of wheat quality properties in flour, dough, and bread should be evaluated in further techno-functional studies of wheat plant materials.

## Figures and Tables

**Figure 1 plants-11-02662-f001:**
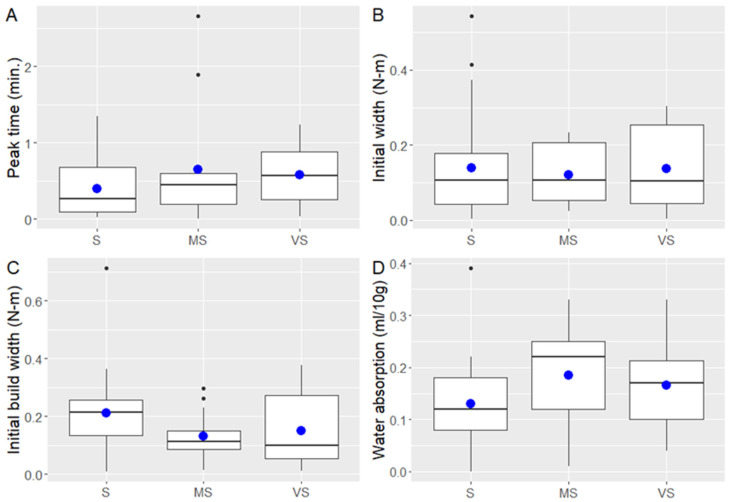
Differences (subtracted between the years) in the dough mixing parameters of 56 spring wheat genotypes grown in the cool (2017) and heat–drought (2018) years: peak time (**A**), initial width (**B**), initial build width (**C**), and water absorption (**D**). Peak time is expressed as minutes (min.), initial width and initial build width are expressed as mixing torque in N·m, water absorption is expressed in ml/10 g of flour. Blue dots represent mean values.

**Figure 2 plants-11-02662-f002:**
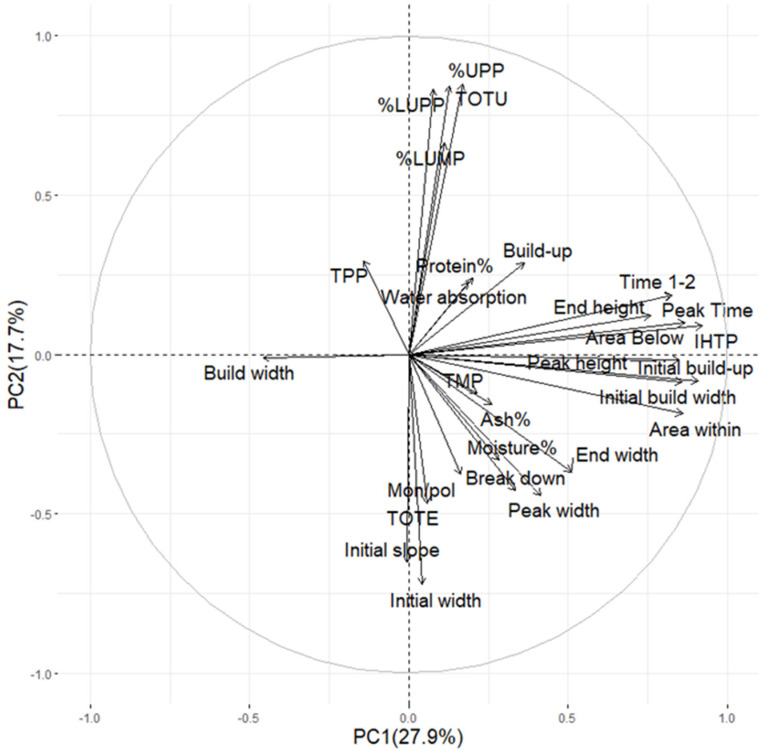
Principal component analysis (PCA) plot of mixograph parameters in dough and protein composition parameters in flour of 56 spring wheat genotypes grown in cool (2017) and heat–drought (2018) years evaluated by SE-HPLC; protein (%), ash (%), and flour moisture (%) were determined in flour by NIT (taken from Lama et al., 2022).

**Figure 3 plants-11-02662-f003:**
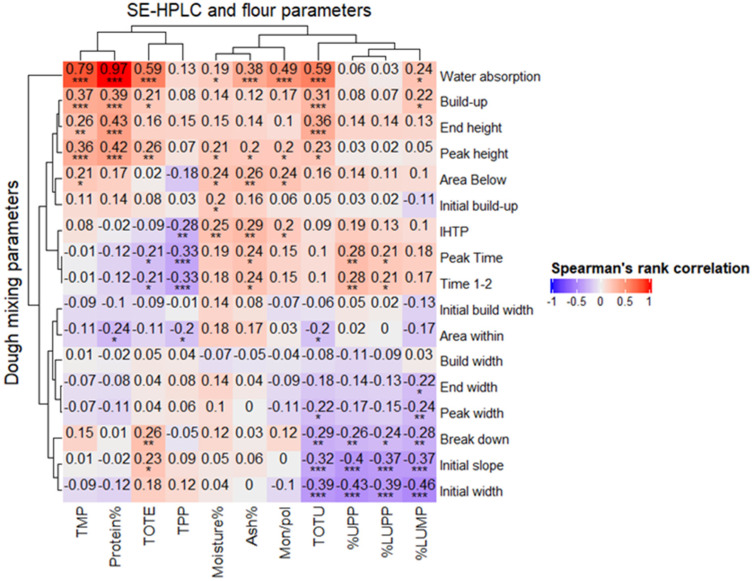
Spearman’s rank correlation matrix and hierarchical clustering of results (dendogram) based on complete linkage method for the gluten protein parameters obtained by SE-HPLC; flour protein (%), moisture (%), and ash (%) determined by NIT (taken from Lama et al. 2022) and 17 dough mixing parameters of 56 spring wheat genotypes grown in cool (2017) and heat–drought (2018) years. ***, **, and * indicate significance at *p* < 0.001, *p* < 0.01, and *p* < 0.05.

**Figure 4 plants-11-02662-f004:**
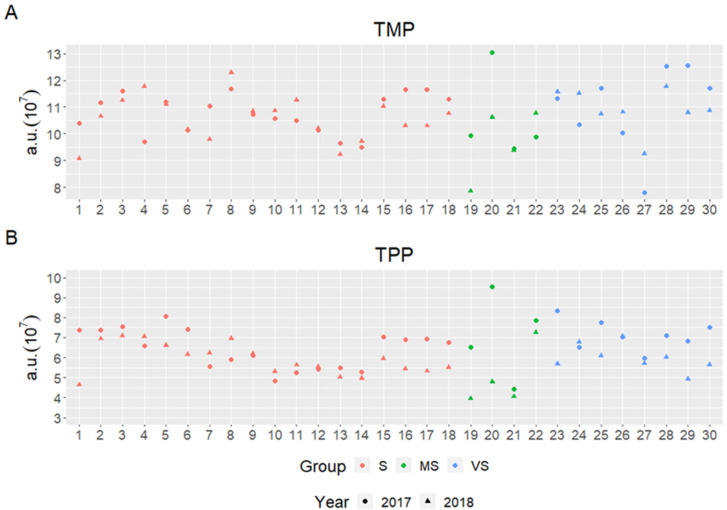
Total amount of monomeric proteins (TMP) (**A**) and total amount of polymeric proteins (TPP) (**B**) in dough of wheat samples shown in Groups S, MS, and VS of 30 spring wheat genotypes grown in cool (2017) and heat–drought (2018) years.

**Figure 5 plants-11-02662-f005:**
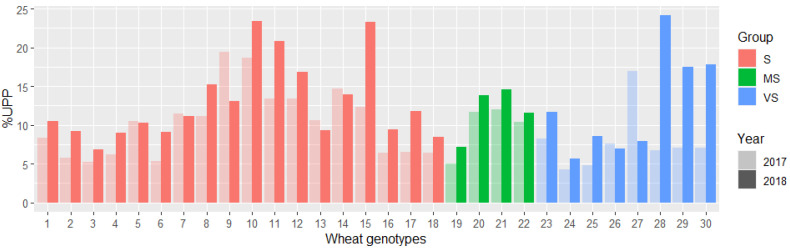
Unextractable polymeric protein (%UPP) in wheat dough samples in Groups S, MS, and VS of 30 spring wheat genotypes grown in cool (2017) and heat–drought (2018) years.

**Figure 6 plants-11-02662-f006:**
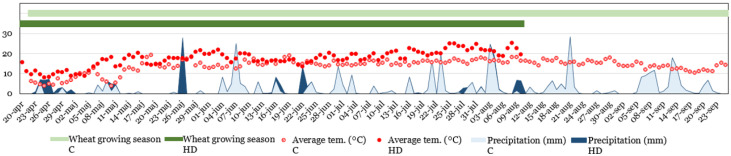
Growing conditions of 56 wheat genotypes grown in two varying environments used in this study; average temperature (°C) and precipitation (mm) during the wheat growing seasons (C—cool, 2017; HT—heat–drought, 2018) expressed as number of days.

**Figure 7 plants-11-02662-f007:**
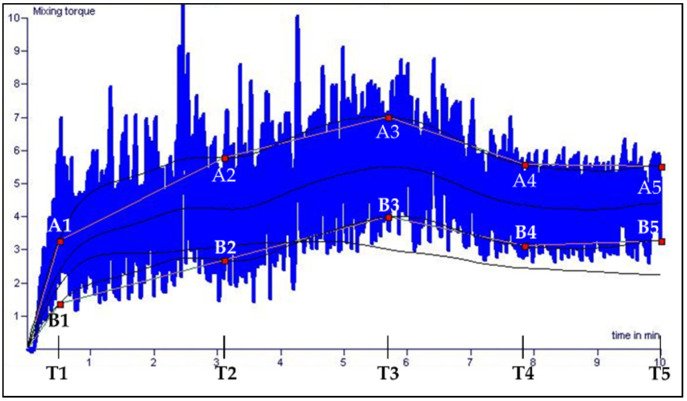
Mixograph curve for wheat genotype Mirakel grown in the cool year (2017).

**Table 1 plants-11-02662-t001:** Analysis of variance (ANOVA) showing the effect of group, year, and wheat genotype and their interaction on the mixing characteristics of the 56 spring wheat genotypes grown in cool (2017) and heat–drought (2018) years. Data are presented as mean values calculated from three replicates for each parameter.

	Group	Year	Group × Genotype	Group × Year	Group × Year × Genotype	Residuals
Df	2	1	53	2	53	224
Water absorption						
Initial slope	0.37 *	4.41 ***	39.92 ***	2.23 ***	18.31 ***	9.75
Initial width	0.05 **	0.58 ***	4.84 ***	0.23 ***	1.79 ***	1.01
Initial build-up	0.45 *	3.52 ***	49.22 ***	1.93 ***	14.24 ***	16.02
Time 1–2	6.30 ***	0.08	53.71 ***	1.32 ***	8.99 ***	4.85
Initial build width	0.07	0.37 ***	19.51 ***	0.45 ***	3.14 ***	4.65
Dough development						
Buildup	0.69	0.05	27.88 ***	1.56 **	13.70 **	31.57
Peak time	25.30 ***	0.32	215.09 ***	5.23 ***	35.94 ***	19.37
Peak height	1.07 *	5.77 ***	70.57 ***	2.78 ***	26.64 ***	28.09
Peak width	1.43 ***	0.35 ***	24.50 ***	0.05	4.24 ***	5
Build width	1.10 ***	0.60 ***	10.31 ***	0.75 ***	6.00 ***	9.63
Dough breakdown						
Breakdown	0.45	0.07	7.33	0.05	5.05	260.240
IHTP	500.4 ***	13.5 *	4830.6 ***	164.8 ***	1017.8 ***	597.0
Water absorption	0.09	0.32***	-	0.12	-	2.41

Note: IHTP—Integrated height to the peak. ***, **, and * indicate significance at *p* < 0.001, *p* < 0.01, and *p* < 0.05, respectively. The mixograph parameters (initial slope, initial width, initial buildup, build width, buildup, peak height, peak width, width build, breakdown, and IHTP) are measured as torque (N·m); mixing time (time 1–2 and peak time) are measured in minutes; water absorption is measured ml/10 g flour.

**Table 2 plants-11-02662-t002:** Tukey’s post hoc test (*p* < 0.05) of dough mixing characteristics of groups (S—stable, MS—moderately stable, and VS—varying stability) of 56 spring wheat genotypes grown in cool (C, 2017) and heat–drought (HD, 2018) years. Data are presented as means ± standard errors, calculated from three replicates for each parameter.

	Group S-C	Group S-HD	Group MS-C	Group MS-HD	Group VS-C	Group VS-HD
Water absorption						
Initial slope	4.53 ± 0.07 ab	4.23 ± 0.07 c	4.33 ± 0.10 abc	4.40 ± 0.10 abc	4.64 ± 0.11 a	4.27 ± 0.11 cb
Initial width	1.23 ± 0.02 ab	1.12 ± 0.02 cd	1.14 ± 0.04 abcd	1.15 ± 0.04 abcd	1.24 ± 0.04 ac	1.12 ± 0.04 bd
Initial buildup	3.47 ± 0.07 a	3.13 ± 0.08 b	3.41 ± 0.12 ab	3.30 ± 0.12 ab	3.23 ± 0.13 ab	3.26 ± 0.13 ab
Time 1–2	1.65 ± 0.08 ab	1.57 ± 0.08 b	1.85 ± 0.12 b	2.05 ± 0.12 a	1.67 ± 0.13 ab	1.81 ± 0.13 ab
Initial build width	1.97 ± 0.05 a	1.84 ± 0.05 b	1.92 ± 0.07 ab	1.89 ± 0.07 ab	1.84 ± 0.07 ab	1.90 ± 0.07 ab
Dough development						
Buildup	1.46 ± 0.06 a	1.50 ± 0.07 a	1.57 ± 0.10 a	1.37 ± 0.09 a	1.48 ± 0.10 a	1.69 ± 0.10 a
Peak time	3.80 ± 0.16 ab	3.65 ± 0.15 b	4.20 ± 0.23 b	4.60 ± 0.25 a	3.85 ± 0.24 ab	4.12 ± 0.24 ab
Peak height	5.50 ± 0.09 a	5.10 ± 0.10 b	5.54 ± 0.14 ab	5.31 ± 0.15 ab	5.38 ± 0.15 ab	5.44 ± 0.16 ab
Peak width	3.37 ± 0.05 a	3.30 ± 0.05 a	3.22 ± 0.08 a	3.13 ± 0.08 a	3.32 ± 0.08 a	3.30 ± 0.08 a
Width build	0.17 ± 0.04 b	0.33 ± 0.04 a	0.15 ± 0.06 ab	0.09 ± 0.06 b	0.24 ± 0.06 ab	0.28 ± 0.06 ab
Breakdown phase						
Breakdown	0.70 ± 0.09 a	0.65 ± 0.09 a	0.59 ± 0.15 a	0.59 ± 0.15 a	0.69 ± 0.15 a	0.69 ± 0.15 a
IHTP	13.8 ± 0.72 a	12.1 ± 0.72 b	15.2 ± 1.12 ab	16.7 ± 1.12 a	13.6 ± 1.16 ab	14.4 ± 1.16 ab
Water absorption	6.60 ± 0.03 a	6.49 ± 0.03 bc	6.63 ± 0.04 ab	6.45 ± 0.04 c	6.62 ± 0.05 abc	6.60 ± 0.04 abc

Note: Different letters (a, b, c and d) indicate significant differences among the groups and years for each parameter. The mixograph parameters (initial slope, initial width, initial buildup, build width, buildup, peak height, peak width, width build, breakdown, and IHTP) are measured as torque (N·m); mixing time (time 1–2 and peak time) is measured in minutes; water absorption is measured in ml/10 g flour.

## Data Availability

The data are openly available from the first author and can be accessed upon reasonable request.

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
