# Peer review of "Striving for Stability in the Dough Mixing Quality of Spring Wheat under the Influence of Prolonged Heat and Drought"

_plants, 2022, doi:10.3390/plants11192662_

Round 1

Reviewer 1 Report

The paper has a very promising starting point: Striving for stable the dough mixing quality in spring wheat 2 under the influence of prolonged heat and drought

Major points:

1. Summaries need to be clear and concise

2. In the instruction section, you should give a update of what is known and unknown

Minor points:

 For Tables: in Table 2, what is the unit of these numbers? Are these statistic results? Where can we see the statistic different? How many replicates? In Table 2, the unit?

In all figures: Missing population number (with clear indications of what it represents) and statistical test used is missing in all figure legends 

Author Response

Response to the reviewer 1 comments on the manuscript ID plants-1943197

We have addressed the reviewers’ comments in a point-by-point fashion, which are specified bellow.

Major points

  1. Summaries need to be clear and concise

We are thankful for the suggestions to improve our manuscript. We substantially modified the abstract and conclusions in order to make them clear and concise.

  1. In the instruction section, you should give a update of what is known and unknown.

The introduction section is updated and improved according to the suggestions and specific information on what is known and unknown is now included and highlighted.

Minor points:

  1. For Tables: in Table 2, what is the unit of these numbers? Are these statistic results? Where can we see the statistic different? How many replicates? In Table 2, the unit?

All the comments have been addressed and units, replicates and other requested info are now included, and the changes are highlighted.

  1. In all figures: Missing population number (with clear indications of what it represents) and statistical test used is missing in all figure legends 

Number of the population is added in all the figures.

Reviewer 2 Report

The manuscript is written well.

A concern in the manuscript is that this manuscript has a significant amount overlapping published data (Lama et al (2022) Food & Energy Security 15th March). It would be nice to remove the data repetition and present the results concisely or indicate clearly that the data taken from the following reference(s).

Elaborate methodologies are given in this manuscript, authors may refer their own publications and present the data concisely.

Supplemental information is not required because this manuscript is an extension of their earlier publication (March 2022).

It would be nice to avoid 'Protein Concentration' in the manuscript and change to Protein Amounts / Content.

Author Response

Response to the reviewer 2 comments on the manuscript ID plants-1943197

We have addressed the reviewers’ comments in a point-by-point fashion, which are specified bellow.

  1. A concern in the manuscript is that this manuscript has a significant amount overlapping published data (Lama et al (2022) Food & Energy Security 15th March). It would be nice to remove the data repetition and present the results concisely or indicate clearly that the data taken from the following reference(s).

We thank the reviewer for the comments, which we addressed carefully in order to improve the manuscript. We removed flour parameters from Table1 and Table 2 and we referred to our previous publication throughout the manuscript. We also clearly indicated that this study is an extension of our previous study and this was highlighted in the Materials and methods section 4.1.

  1. Elaborate methodologies are given in this manuscript, authors may refer their own publications and present the data concisely.

We addressed the comment and referred to our own publications in all sections of the methodology, such as 4.1, 4.2. and 4.4. All the major changes are highlighted.

  1. Supplemental information is not required because this manuscript is an extension of their earlier publication (March 2022).

The supplemental information has been removed. 

  1. It would be nice to avoid 'Protein Concentration' in the manuscript and change to Protein Amounts / Content.

Protein concentration has been replaced with protein content throughout the manuscript.
